# Effect of FeSi Alloy Additions and Calcium Treatment on Non-Metallic Inclusions in 304 Stainless Steel during AOD and LF Refining Process

**Jun Zhai [1,2,*], Chengbin Shi [3,*], Weiyun Lang [1,2], Yu Zhao [3] and Shijun Wang [3]**

[1]  State Key Laboratory of Advanced Stainless Steel Materials, Taiyuan Iron & Steel (Group) Co., Ltd., Taiyuan 030003, China
[2]  Technology Center, Shanxi Taigang Stainless Steel Co., Ltd., Taiyuan 030003, China
[3]  State Key Laboratory of Advanced Metallurgy, University of Science and Technology Beijing, Beijing 100083, China
*  Correspondence: zhaijun01@tisco.com.cn (J.Z.); chengbin.shi@ustb.edu.cn (C.S.)

**Abstract:** Non-deformable inclusions are detrimental to the surface quality and mechanical properties of stainless-steel plates. Plant trials were conducted to investigate the effect of different ferrosilicon alloys and calcium treatment during argon oxygen decarburization (AOD) and ladle furnace (LF) refining on inclusions in Si-killed 304 stainless steel. The inclusions were examined by scanning electron microscope with energy dispersive spectrometer. The results show that both the contents of soluble aluminum in molten steel and $Al_2O_3$ in slag increase with the increase of aluminum content in FeSi alloy. The content of soluble aluminum in liquid steel could be limited to lower than 0.004% when using ultra-purity FeSi alloy. When the calcium wire addition is 2 m/t, inclusions are located in the low-melting-temperature region, and the inclusion rating of hot rolled plates is mainly C-class. Industrial application shows that, by decreasing the soluble aluminum content in liquid steel, decreasing the MgO and $Al_2O_3$ in slag in AOD, and applying low basicity refining slag as well as calcium treatment, the inclusions are low melting point silicates. The inclusion rating of hot rolled plates is mainly fine C-type with a small amount of class-A, and surface polishing qualification rate is increased from 17.8% to more than 88.7%.

**Keywords:** inclusions; stainless steel; slag; calcium treatment; LF refining

## 1. Introduction

Non-metallic inclusions with high-melting points and strength, such as $Al_2O_3$ and $MgAl_2O_4$, are detrimental to the surface quality and mechanical properties of stainless steel. The detriments of non-metallic inclusions to the steel properties have been summarized in the review articles by Park et al. [1] for stainless steel. Great efforts have been put forward to minimize the number of inclusions by decreasing the oxygen content in steel during the steelmaking process. Many researchers have investigated the evolution of inclusions and their affecting factors during steelmaking processing of stainless steel [2–7]. The low-melting temperature inclusions had a desirable chemical composition for plastic behavior in subsequent processing and service operations. Therefore, targeting oxide inclusion compositions with low-melting temperature is an important aspect of inclusion control in stainless steel [8–10].

Slag composition is one of most important factors in determining the compositions of oxide inclusions in stainless steel [11–14]. Kim et al. [15] studied the effect of tundish flux on compositional changes in non-metallic inclusions in stainless steel, and found that the aluminum originating from the slag modifies the preexisting Mn-silicate inclusions into alumina-rich inclusions in the steel. The study by the present authors [16] shows that the MgO and $Al_2O_3$ content of inclusions in slab increase by 14.4% and 9.1% respectively with

increasing slag basicity from 1.5 to 2.6 in LF refining process, and there are no MgO·$Al_2O_3$ spinel inclusions in continuous casting slab when slag basicity is less than 1.9 in LF refining. $Al_2O_3$ and $MgAl_2O_4$ spinel inclusions are frequently observed in stainless steel. Calcium treatment is an effective way to modify compositions of oxide inclusions to low-melting-temperature state [17–20]. Du et al. [19] reported that most of the inclusions in 316 L Si-killed stainless steel without calcium treatment were MnS and MnO-$Al_2O_3$-MgO-$TiO_x$ inclusions, whereas the inclusions were mainly spherical CaO-$SiO_2$-MgO-$Al_2O_3$-MnS inclusions in calcium-treated 316 Si-killed stainless steel. Wang et al. [21] found that calcium treatment was necessary to modify the ferrosilicon deoxidation product into low melting temperature inclusions in the low-Al region (from 40 to 100 ppm) using CaO-$SiO_2$-$Al_2O_3$ slag, and calcium addition has a "liquid window" where adding calcium can accelerate inclusion modification. Wang et al. [22] stated that the MnO-$SiO_2$-$Al_2O_3$-based inclusions were difficult to change into CaO-$SiO_2$-$Al_2O_3$-based inclusions, because the dissolved Ca in liquid steel was not high enough.

The addition of ferroalloys could result in a supply of additional inclusions in the steel [23,24]. The effect of ferrosilicon additions on the composition of inclusions has been studied by several researchers [25–29]. Park and Kang [25] reported that the addition of FeSi with a higher Al content induced an increase in $Al_2O_3$ content in the inclusions and promoted formation of spinel inclusions. Wijk and Brabie [26] studied the influence of FeSi75 alloy additions on the inclusions, and found that silica inclusions were formed after the addition of a high-purity FeSi75 alloy (0.004 mass pct Al, 0.006 mass% Ca, 0.016 mass% O). Mizuno et al. [27] reported that the presence of 1.7 mass% Ca in FeSi alloys prevented the formation of spinel inclusions. Li et al. [28] reported that the high Al (1.6 mass pct) containing FeSi alloy leads to a significantly increased $Al_2O_3$ content in $Al_2O_3$–$SiO_2$–MnO inclusions and to the formation of pure $Al_2O_3$ inclusions in stainless steel. Recent work by Wang et al. [29] shows that $MnCr_2O_4$ spinel inclusions originating from the FeCr alloys transformed into $Ti_2O_3$-$Cr_2O_3$-based liquid inclusions and $Ti_2O_3$-rich solid inclusions due to the reactions between $MnCr_2O_4$ and TiN inclusions or dissolved Ti in liquid steel. Therefore, it is quietly necessary to determine the impurities in ferroalloys to meet the increasing demand for strict inclusion control.

To avoid hard alumina and $MgAl_2O_4$ spinel inclusions resulting from Al deoxidation, Mn-Si deoxidation is used in stainless steel steelmaking. The present study was conducted with the aim to reveal the effects of ferrosilicon alloy with different aluminum contents and calcium treatment on inclusions in 304 stainless steel based on industrial-scale trials. In order to clarify the evolution of inclusions in Si-killed stainless steel, industrial samples were taken during the industrial argon oxygen decarburization and ladle refining of 304 stainless steel.

## 2. Experimental

### 2.1. Plaint Trials

The plant trials were conducted in the production route of 160 t (EAF) → 180 t (AOD) → ladle furnace (LF) → continuous casting (CC) for producing 304 stainless steel. FeSi alloys differ in their aluminum contents owing to their different manufacturing routes. Different grades of FeSi alloys are selected for individual AOD process, i.e., ordinary ferrosilicon, low-Al ferrosilicon, and ultra-purity ferrosilicon alloys. The chemical compositions of ferrosilicon alloys are shown in Table 1. Three AOD plant trials were conducted. Different grades of FeSi alloys were added at the reduction stage of AOD refining process. The addition amount of FeSi alloys is 22.6 kg/t, 26.8 kg/t and 25.3 kg/t, respectively. Steel and slag samples were taken at the final stage of AOD refining. In addition, steel samples were taken from casting slab and hot rolled plate.

**Table 1.** Chemical compositions of ferrosilicon alloys (mass%).

|  | Si | Al | Ca | Mn | S | P | Fe |
|---|---|---|---|---|---|---|---|
| ordinary FeSi alloy | 74.63 | 2.5 | 0.505 | 0.11 | 0.003 | 0.017 | Bal. |
| low-Al FeSi alloy | 74.91 | 0.8 | 0.286 | 0.15 | 0.002 | 0.022 | Bal. |
| ultra-purity FeSi alloy | 77.43 | 0.2 | 0.028 | 0.10 | 0.002 | 0.013 | Bal. |

Calcium treatment was employed aiming to increase the CaO content in oxide inclusions, eventually bringing the compositions of inclusions in low melting point range. In order to study the effect of calcium addition amount on inclusions, Si-Ca wires (70 mass% Si–30 mass% Ca) were added before the end of LF refining. In the case of calcium treatment plant trials, the ultra-purity FeSi alloy is used at the reduction stage of AOD refining process and the slag with basicity of 1.5 is used in LF refining.

The main manufacturing steps, and steel and slag sampling are presented in Figure 1. Steel and slag samples were taken at the arrival, before calcium addition, and final stage of LF refining. In addition, steel samples were taken from casting slab and hot rolled plate.

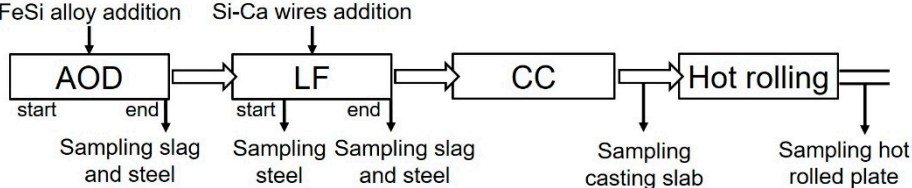

**Figure 1.** Schematic layout of the steelmaking process and sampling occasions.

### 2.2. Chemical Analysis and Microscopic Observation

The steel samples were prepared for chemical analysis. The aluminum content of steel samples was measured by inductively coupled plasma atomic emission spectroscopy (ICP-AES). The total oxygen and sulfur contents in the steel were measured by the inert gas fusion-infrared absorptiometry and the combustion-infrared absorption method, respectively. The nitrogen content was determined by the inert gas fusion-thermal conductivity method. The chemical composition of refining slag was analyzed using an X-ray fluorescence spectrometer (XRF).

The cross sections of the steel samples were prepared for metallographic analysis by progressive grinding using SiC paper and polishing with 3 and 1 μm diamond paste. The polished surfaces were observed by an automatic scanning electron microscopy (SEM) with energy-dispersive X-ray spectroscopy (EDS) named ASPEX automatic scanning electron microscope (FEI, Hillsboro, OR, USA). The Chinese standard GB/T 10561-2005 was used to grade the inclusions in the hot plate samples.

### 3. Results and Discussion

#### 3.1. Effect of Different Grades of FeSi Addition at Reduction Stage of AOD on Inclusions

The chemical composition of liquid steel in trials is shown in Table 2. It can be seen from the Table 2 that the aluminum content in oxidation stage of AOD is lower than 30 ppm. The content of Al reduced by AOD is 63 ppm with ordinary FeSi alloy, and the corresponding Al content is 41 ppm when low-Al FeSi alloy is used, while the content of Al is 32 ppm using ultra-purity FeSi alloy. In addition, Al content tends to decrease in LF refining process.

**Table 2.** Chemical composition of liquid steel (mass%).

|  | Stage | C | Si | Mn | P | S | Cr | Ni | Al | N |
|---|---|---|---|---|---|---|---|---|---|---|
| Ordinary FeSi alloy | AOD oxidation stage | 0.144 | 0.00 | 0.13 | 0.031 | 0.0880 | 17.83 | 7.83 | 0.0026 | 0.086 |
|  | AOD final stage | 0.045 | 0.66 | 1.14 | 0.033 | 0.0016 | 18.18 | 7.98 | 0.0063 | 0.046 |
|  | LF final stage | 0.046 | 0.65 | 1.15 | 0.033 | 0.0012 | 18.00 | 7.96 | 0.0057 | 0.047 |
| low-Al FeSi alloy | AOD oxidation stage | 0.142 | 0.00 | 0.12 | 0.030 | 0.0813 | 17.52 | 8.09 | 0.0026 | 0.102 |
|  | AOD final stage | 0.044 | 0.55 | 1.15 | 0.032 | 0.0024 | 18.01 | 8.01 | 0.0041 | 0.042 |
|  | LF final stage | 0.043 | 0.53 | 1.14 | 0.032 | 0.0019 | 18.20 | 7.98 | 0.0037 | 0.042 |
| ultra-purity FeSi alloy | AOD oxidation stage | 0.112 | 0.00 | 0.11 | 0.028 | 0.1125 | 17.50 | 7.89 | 0.0025 | 0.094 |
|  | AOD final stage | 0.044 | 0.49 | 1.11 | 0.030 | 0.0037 | 18.12 | 7.88 | 0.0032 | 0.039 |
|  | LF final stage | 0.045 | 0.48 | 1.11 | 0.030 | 0.0017 | 18.09 | 7.98 | 0.0032 | 0.041 |

The chemical composition of slag at different stages is given in Table 3. As can be seen, the basicity of slag is controlled between 2.0–2.2 during AOD tapping, and the $Al_2O_3$ content in slag can reach more than 3.0% using ordinary FeSi alloy, while the $Al_2O_3$ content is no more than 2.0% when low-Al FeSi alloy and ultra pure FeSi alloy are used. The basicity of slag is controlled between 2.4 and 2.6 after LF slag adjustment, and it tends to increase during LF treatment.

**Table 3.** Chemical composition of refining slag during the test (mass%).

|  | Stage | CaO | $SiO_2$ | $Al_2O_3$ | MgO | MnO | FeO | $Cr_2O_3$ | S | $CaF_2$ | $R$ ($CaO/SiO_2$) |
|---|---|---|---|---|---|---|---|---|---|---|---|
| Ordinary FeSi alloy | AOD final stage | 58.5 | 27 | 3.2 | 8.1 | 0.19 | 0.65 | 0.11 | 0.22 | 5.1 | 2.17 |
|  | LF final stage | 58.4 | 24.6 | 2.9 | 7 | 0.18 | 0.29 | 0.09 | 0.34 | 5.2 | 2.37 |
| low-Al FeSi alloy | AOD final stage | 55.6 | 27.0 | 2.0 | 9.7 | 0.21 | 0.51 | 0.16 | 0.15 | 5.7 | 2.06 |
|  | LF final stage | 62.4 | 25.1 | 1.6 | 5.9 | 0.20 | 0.31 | 0.11 | 0.26 | 8.0 | 2.49 |
| ultra-purity FeSi alloy | AOD final stage | 60.7 | 28.0 | 1.7 | 7.8 | 0.24 | 0.90 | 0.34 | 0.76 | 5.2 | 2.17 |
|  | LF final stage | 63.4 | 25.7 | 1.5 | 8.5 | 0.19 | 0.53 | 0.03 | 0.32 | 6.5 | 2.46 |

### 3.1.1. Content Changes of Al in Molten Steel and $Al_2O_3$ in Slag

Figure 2 shows the influence of different ferrosilicon alloy types on Al content in liquid steel, yield of aluminum and $Al_2O_3$ in slag. The yield of aluminum is calculated as follows:

$$\text{Yield of aluminum} = \frac{\left([Al]_f - [Al]_i\right) \times 1000}{[Al]_{\text{alloy}} \times m_{\text{alloy}}}$$

where $[Al]_i$ is the Al content in liquid steel before ferrosilicon addition.

$[Al]_f$ is the Al content in liquid steel after ferrosilicon addition.

$[Al]_{\text{alloy}}$ is the Al content in ferrosilicon.

$m_{\text{alloy}}$ is the ferrosilicon addition amount per ton of liquid steel.

It can be observed that the Al content in molten steel after AOD tapping with ordinary FeSi alloy, low-Al FeSi alloy and ultra-pure FeSi alloy is 57 ppm, 37 ppm and 32 ppm respectively, and the corresponding yield of aluminum is 6.5%, 7.9% and 14% respectively, indicating that most of the aluminum entering liquid steel in ferrosilicon alloy participates in deoxidation and enters into slag as inclusions. The $Al_2O_3$ content of the slag corresponding to three ferrosilicon alloys is 3.2 mass%, 2.0 mass% and 1.7 mass%, respectively.

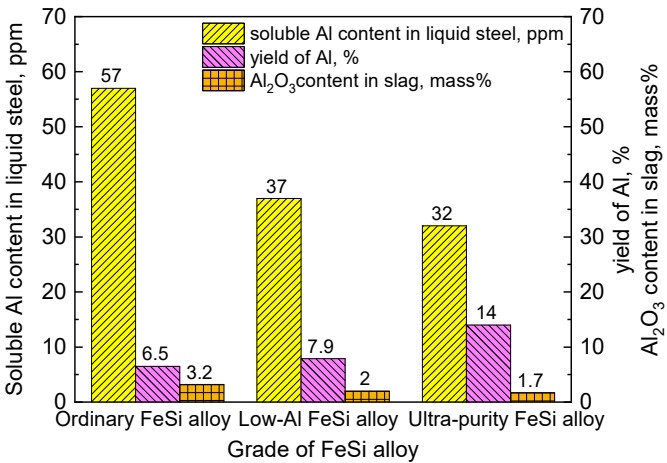

**Figure 2.** Comparison of Al content in molten steel and $Al_2O_3$ content in slag when using different ferrosilicon alloys.

### 3.1.2. Effect of Ferrosilicon Alloy Types on Inclusion Types

Figures 3–5 show the morphology and EDS spectra of typical inclusions corresponding to three ferrosilicon alloys during AOD tapping. As can be seen, the peak value of Al is similar to that of Ca and Si in inclusion energy spectrum when ordinary FeSi alloy is used. The peak value of Al decreases significantly using low-Al FeSi alloy and ultra-pure FeSi alloy.

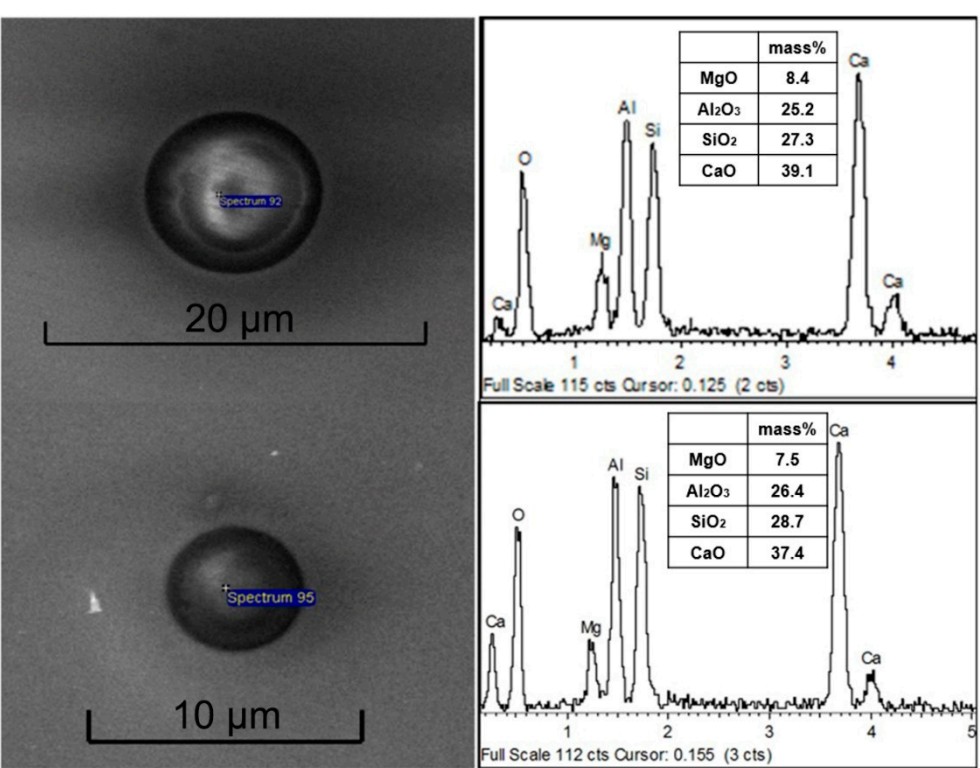

**Figure 3.** SEM images and EDS results of inclusions in the samples taken at AOD tapping when using ordinary FeSi alloy.

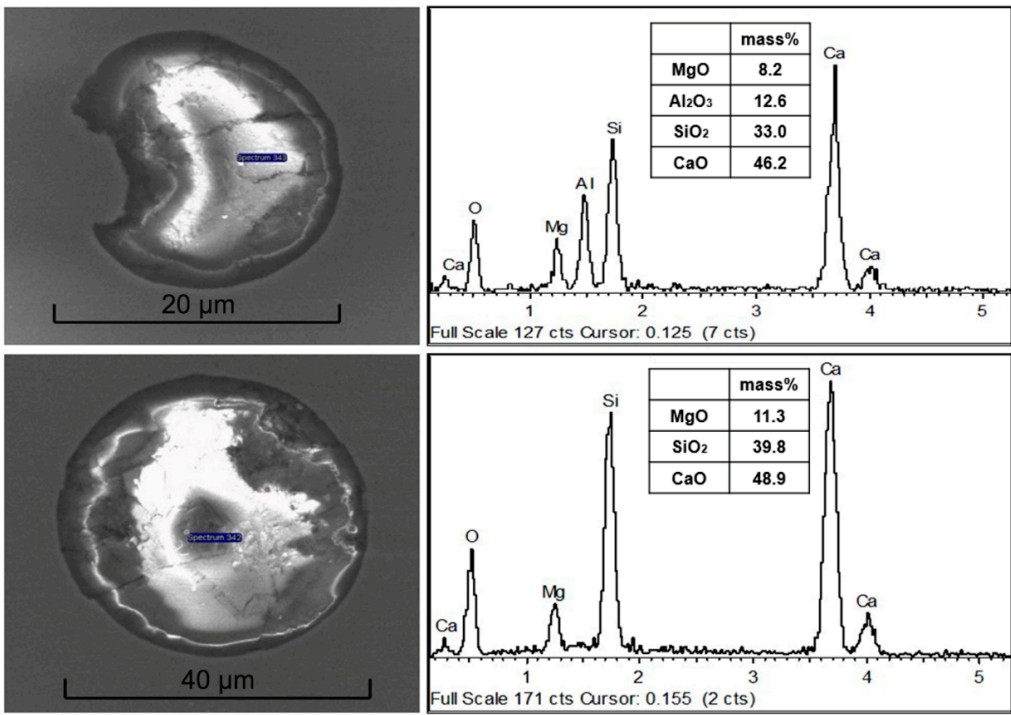

**Figure 4.** SEM images and EDS results of inclusions in the samples taken at AOD tapping when using low-Al FeSi alloy.

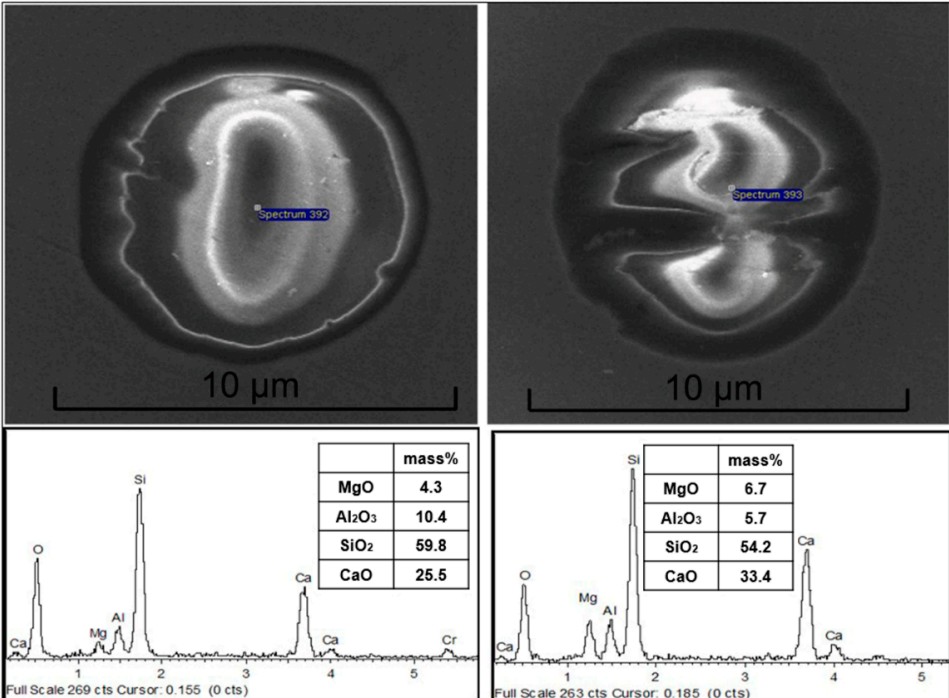

**Figure 5.** SEM images and EDS results of inclusions in the samples taken at AOD tapping when using ultra-purity FeSi alloy.

The composition distribution of inclusion in the samples taken at AOD tapping when using three ferrosilicon alloys on $CaO-SiO_2-Al_2O_3$-5 mass% MgO phase diagram is shown in Figure 6. In the analysis process, the composition of Cr, Mn and Fe in inclusions is not considered. In the phase diagram, the MgO content is 5 mass%. It can be observed from Figure 6 that the $Al_2O_3$ content in the inclusion can reach 40 mass% when ordinary FeSi

alloy is used, the $Al_2O_3$ content with low-Al FeSi alloy can reach more than 20 mass%, and the $Al_2O_3$ content using ultra-pure FeSi alloy is about 15%, indicating that the $Al_2O_3$ content in the inclusion increases with the increase of Al content in ferrosilicon alloy. Therefore, it can be determined that the type of ferrosilicon alloys has a significant influence on the type of inclusions.

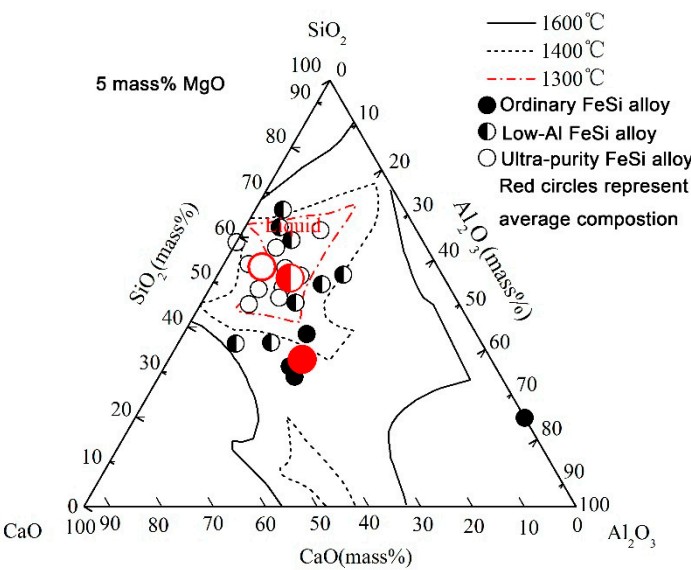

**Figure 6.** The composition distribution of inclusion in the samples taken at AOD tapping when using three ferrosilicon alloys on $CaO$-$SiO_2$-$Al_2O_3$-5 mass% MgO phase diagram.

### 3.1.3. Inclusion Control of Hot Rolled Plate

In order to further determine the influence of ferrosilicon alloy types on the morphology of inclusions, the Chinese standard GB/T 10561-2005 was used to grade the inclusions in the hot plate samples of the test furnaces. The number of samples was 12, and the rating results are shown in Table 4.

**Table 4.** Rating results of inclusion in hot rolled plate corresponding to different ferrosilicon alloys.

|  | A-Type | B-Type | C-Type | D-Type |
|---|---|---|---|---|
| ordinary FeSi alloy | 0 | 0.5~1.0 | 0.5 | 1.0 |
| low-Al FeSi alloy | 0 | 0.5 | 0.5~1.0 | 0.5 |
| ultra-purity FeSi alloy | 0 | 0.5 | 0.5~1.0 | 0.5 |

As can be seen from the table, the inclusions in hot rolled coil are basically the same, and the B-type, C-type and D-type inclusions are predominant. The rating of C-type inclusions has a certain increase trend with low-Al FeSi alloy and ultra-pure FeSi alloy.

Figures 7–9 show the morphology and EDS spectra of typical inclusions in hot rolled coil corresponding to the test furnaces. As illustrated in Figures 7 and 8, magnesia-alumina spinel inclusions were found in the furnaces with ordinary FeSi alloy and low-Al FeSi alloy. No magnesia-alumina spinel inclusions were found in the trial when using ultra-pure FeSi alloy, as shown in Figure 9. However, the MgO and $Al_2O_3$ contents in silicate inclusions increased, indicating that the type of inclusions changed greatly from AOD to finished products. The slag alkalinity increased by about 0.2 during the process from AOD tapping to LF refining. With the increase of slag alkalinity, the MgO and $Al_2O_3$ contents in inclusions continued to increase, and magnesia-alumina spinel inclusions precipitated during solidification.

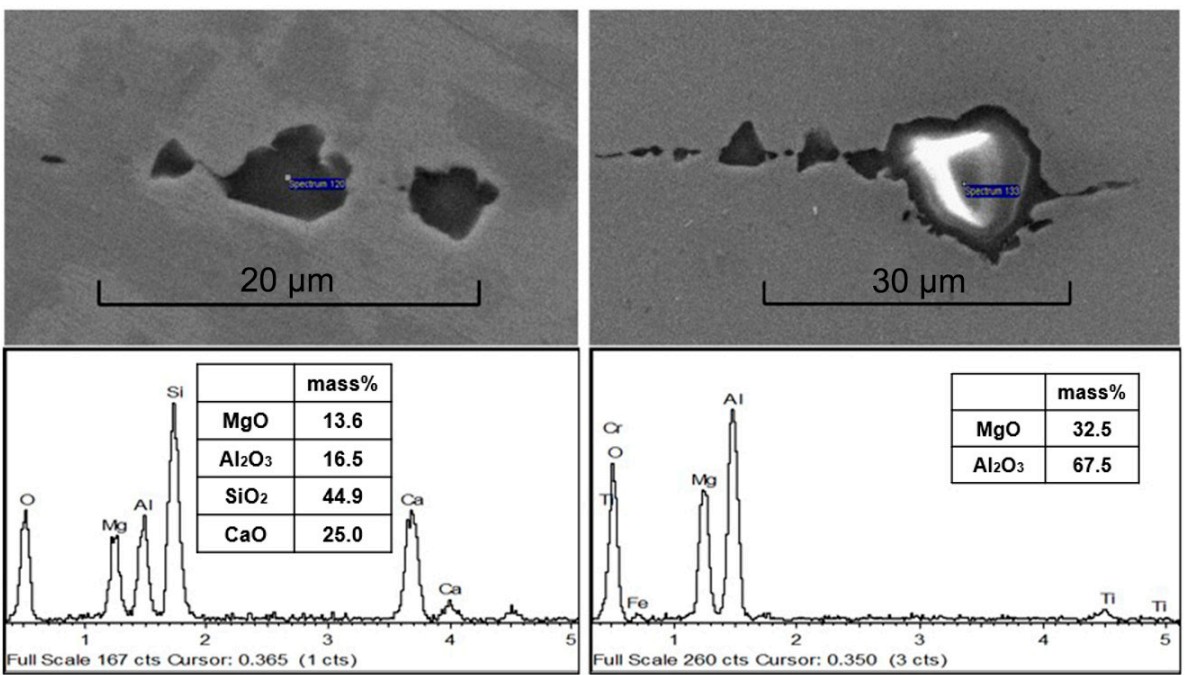

**Figure 7.** SEM images and EDS results of inclusions in hot rolled plates when using ordinary FeSi alloy during AOD reduction.

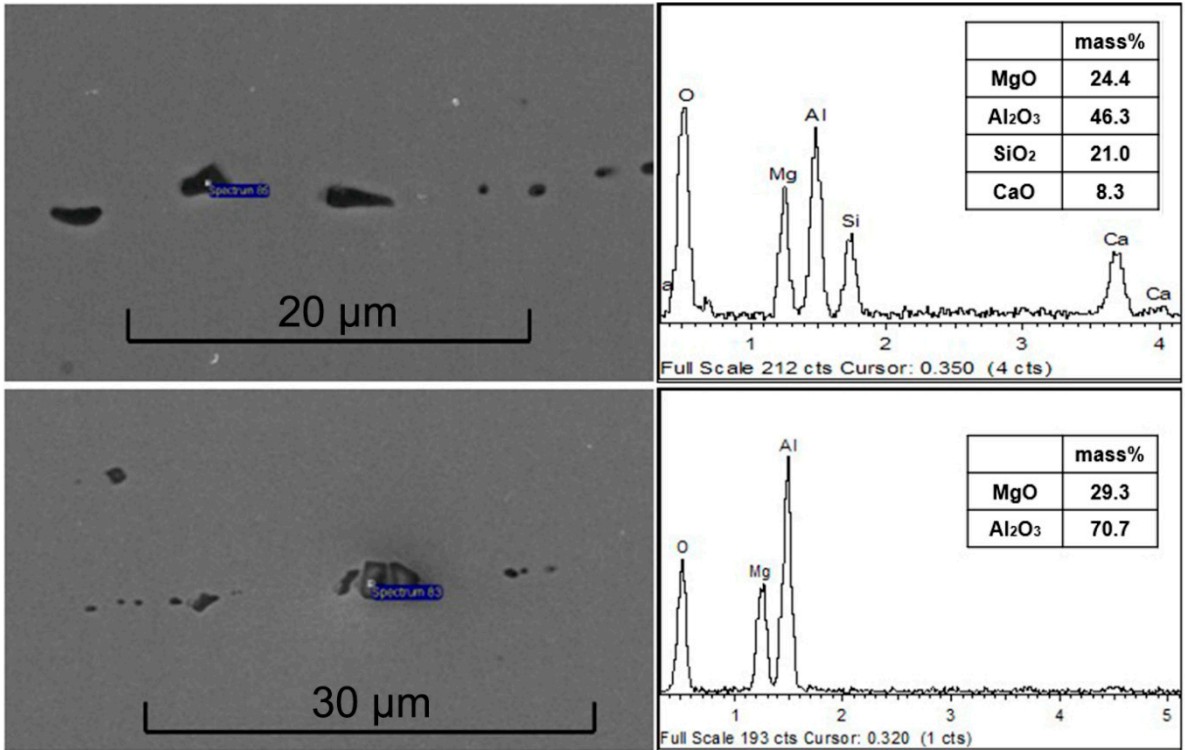

**Figure 8.** SEM images and EDS results of inclusions in hot rolled plates when using low-Al FeSi alloy during AOD reduction.

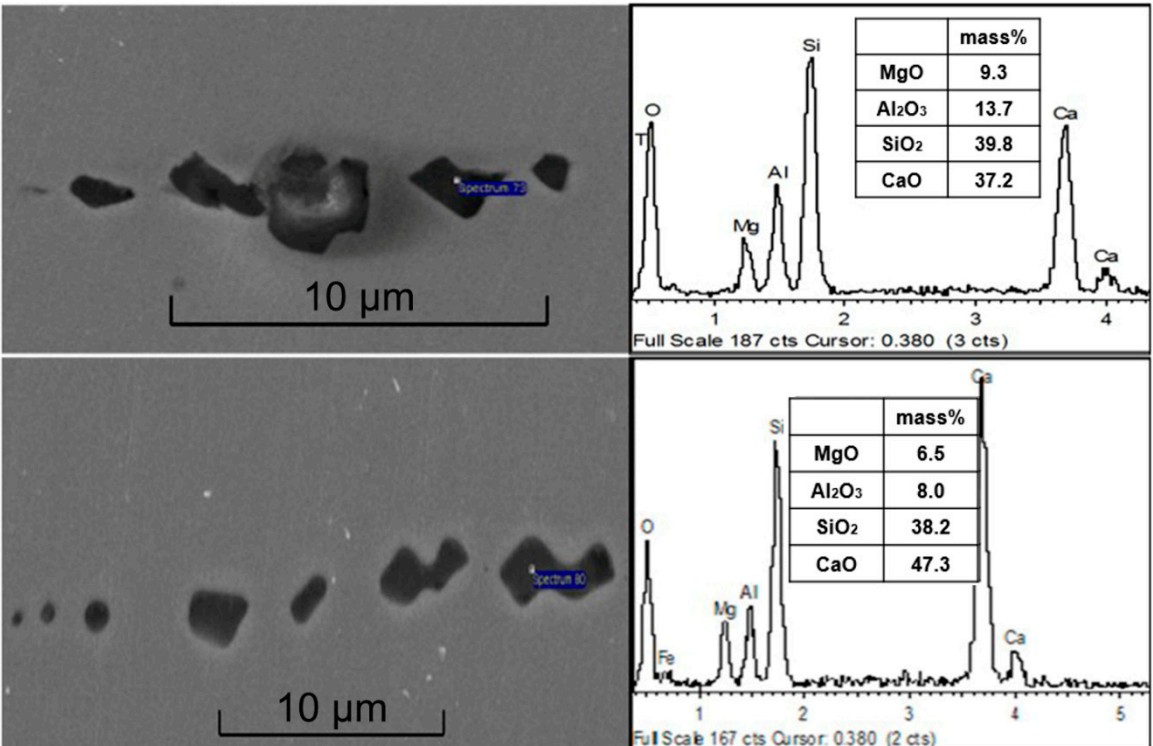

**Figure 9.** SEM images and EDS results of inclusions in hot rolled plates when using ultra-purity FeSi alloy during AOD reduction.

The results of AOD using different ferrosilicon alloys show that using ultra-purity FeSi alloy can reduce the aluminum content in molten steel, magnesia-alumina spinel inclusions and the MgO and $Al_2O_3$ contents in compound magnesia-alumina spinel. In order to reduce $Al_2O_3$ content in inclusion, ultra-purity FeSi alloy was used in subsequent tests.

### 3.2. Effect of Calcium Treatment on Inclusions

When ultra-purity FeSi alloy is used in AOD and the slag with basicity of 1.5 is used in LF, the content of MgO and $Al_2O_3$ in the inclusions is low, $SiO_2$ content is increased, and c-type inclusions in hot rolled plate are increased. Therefore, calcium treatment is added in the test before the end of LF to reduce the content of $SiO_2$ in the inclusions and inhibit the temperature drop of molten steel in the continuous casting process. The increase of MgO and $Al_2O_3$ content in the inclusions eventually makes the inclusions enter the silicate range with low melting point.

At the beginning of LF refining, silica sand is added to control the basicity of refining slag at 1.5–1.6. 10 min before LF weak stirring, the silicon calcium wires of 1 m/t, 2 m/t and 3 m/t were respectively fed according to the scheme design. Table 5 shows the chemical composition of finished steel products of furnace steel.

**Table 5.** Composition of finished steel sample in the tests where different amounts of CaSi wires addition (mass%).

| Heats | Scheme | C | Si | Mn | S | Cr | Ni | Al | Ca |
|---|---|---|---|---|---|---|---|---|---|
| A1600472 | 1 m/t | 0.02 | 0.40 | 1.11 | 0.0038 | 18.3 | 8.10 | 0.0032 | <0.0005 |
| A2602732 | 2 m/t | 0.035 | 0.47 | 1.20 | 0.0022 | 18.3 | 8.15 | 0.0038 | 0.0015 |
| A2600937 | 3 m/t | 0.027 | 0.4 | 1.16 | 0.0019 | 18.2 | 8.13 | 0.0035 | 0.0026 |

Table 6 shows the chemical composition of the slag at end of LF refining in the case of three different amounts of CaSi wires addition in LF refining. It can be seen from the Table 6 that the basicity of LF final slag is basically consistent with the designed basicity.

**Table 6.** Chemical composition of slag at end of LF refining (mass%).

| Heats | Scheme | CaO | SiO$_2$ | Al$_2$O$_3$ | MgO | MnO | Cr$_2$O$_3$ | T.Fe | R (CaO/SiO$_2$) |
|---|---|---|---|---|---|---|---|---|---|
| A1600472 | 1 m/t | 51.3 | 32.9 | 2.5 | 9.5 | 0.93 | 0.76 | 0.37 | 1.56 |
| A2602732 | 2 m/t | 53.7 | 33.5 | 2.3 | 7.6 | 0.7 | 0.57 | 0.27 | 1.60 |
| A2600937 | 3 m/t | 53.0 | 34.8 | 2.7 | 7.9 | 0.84 | 0.69 | 0.30 | 1.52 |

The inclusions in 304 stainless steel are mainly composed of SiO$_2$, Al$_2$O$_3$, MgO and CaO. Therefore, Cr, Mn and Ti are not considered in the statistics of inclusion composition. Table 7 shows the average inclusions in each process in the test process.

**Table 7.** Average compositions of inclusions at each stage of tests with different Si-Ca wires addition (mass%).

| Testing Program | Process | MgO | Al$_2$O$_3$ | SiO$_2$ | CaO |
|---|---|---|---|---|---|
| 1 m/t | beginning of LF | 21.2 | 42.1 | 28.5 | 8.1 |
| | before calcium addition | 3.1 | 14.2 | 79 | 3.7 |
| | End of LF | 1.5 | 12.8 | 72.2 | 13.0 |
| | casting slab | 2.8 | 16.1 | 66.6 | 14.5 |
| 2 m/t | beginning of LF | 18.7 | 36.3 | 35.5 | 9.5 |
| | before calcium addition | 2.9 | 13.2 | 79.6 | 4.3 |
| | End of LF | 0.8 | 11.2 | 60.1 | 27.9 |
| | casting slab | 0.4 | 12.7 | 56.8 | 30.1 |
| 3 m/t | beginning of LF | 22.3 | 39.6 | 26.2 | 11.9 |
| | before calcium addition | 3.7 | 13.5 | 79.7 | 3.1 |
| | end of LF | 0.2 | 9.6 | 45.9 | 44.3 |
| | casting slab | 0.4 | 10.3 | 40.6 | 48.7 |

The size of the inclusions ranged from 5–10 μm, 10–15 μm, 15–20 μm and 20–30 μm. ASPEX automatic analysis results showed that when the size of the inclusions was ≥30 μm, there was a large amount of F element in the composition of the inclusions. It could be determined that these inclusions were originated from slag entrapment in the liquid steel. Therefore, the inclusion with diameter ≥30 μm was removed from inclusion composition, number, and size statistics. The number and size of inclusions in each process of the test furnace are shown in Table 8. It can be seen that the inclusions are mainly in the range of 5–10 μm, and the inclusions larger than 15 μm take up only a small amount.

3.2.1. Effect of Calcium Content on Inclusion Composition

When the basicity of the refining slag system is 1.5–1.6, the content of dissolved oxygen in the molten steel is high. Therefore, the calcium in the feed Si-Ca wires reacts with dissolved oxygen preferentially, and the remaining calcium will react with the silicate inclusions formed. The reactions are shown as follows:

$$[Ca] + [O] = (CaO) \tag{1}$$

$$2[Ca] + (SiO_2) = 2(CaO) + [Si] \tag{2}$$

$$3[Ca] + (Al_2O_3) = 3(CaO) + 2[Al] \tag{3}$$

$$[Ca] + (MgO) = (CaO) + [Mg] \tag{4}$$

**Table 8.** Number and size distribution of inclusions in each process (#/mm$^2$).

| Testing Program | Process | 5–10 μm | 10–15 μm | 15–20 μm | 20–30 μm | Total Number |
|---|---|---|---|---|---|---|
| 1 m/t | beginning of LF | 1.68 | 0.30 | 0.18 | 0.18 | 2.34 |
| | before calcium addition | 0.48 | 0.59 | 0.06 | 0.06 | 0.39 |
| | end of LF | 0.94 | 0.15 | 0.07 | 0.07 | 1.23 |
| | casting slab | 2.78 | 0.57 | 0.14 | 0.14 | 3.63 |
| 2 m/t | beginning of LF | 1.47 | 0.47 | 0.13 | 0.20 | 2.27 |
| | before calcium addition | 0.57 | 0.63 | 0.13 | 0.13 | 0.9 |
| | end of LF | 1.32 | 0.20 | 0.07 | 0.15 | 1.59 |
| | casting slab | 2.73 | 0.37 | 0.07 | 0.12 | 3.17 |
| 3 m/t | beginning of LF | 1.15 | 0.31 | 0.04 | 0.07 | 1.57 |
| | before calcium addition | 0.70 | 0.82 | 0.10 | 0.06 | 1.02 |
| | end of LF | 1.61 | 0.41 | 0.16 | 0.13 | 2.31 |
| | casting slab | 2.97 | 0.37 | 0.01 | 0.05 | 3.4 |

Figure 10 shows the influence of calcium treatment amount on the content of MgO and Al$_2$O$_3$ in the inclusions.

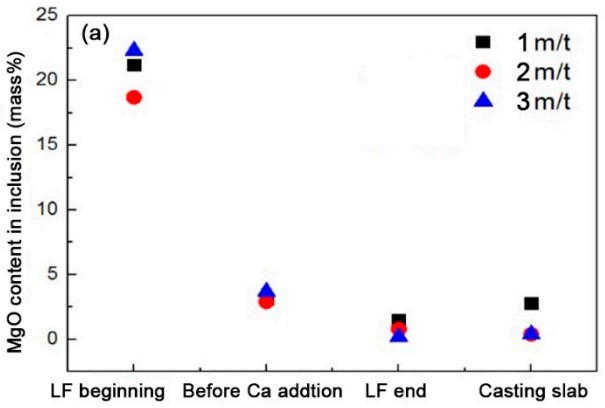 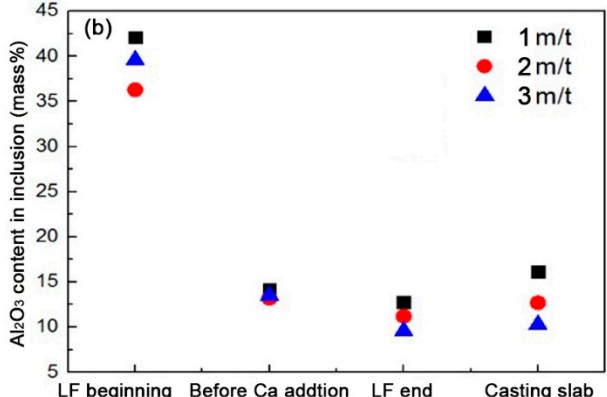

**Figure 10.** Influence of Si-Ca addition amount on MgO and Al$_2$O$_3$ contents of inclusions at different stages. (**a**) MgO content, (**b**) Al$_2$O$_3$ content.

With the addition of Si-Ca wire at the later stage of LF refining, the content of calcium in molten steel increases, while the content of MgO in the inclusions further decreases from about 7% to about 2%. With the increase of Si-Ca wire addition, the content of MgO decreases. After the addition of Si-Ca line, the content of Al$_2$O$_3$ in the inclusion also decreased from 16% to 10%. With the increase of Si-Ca line addition, the amount of Al$_2$O$_3$ content decreased also tended to decrease.

It can be seen from Figure 10 that from end of LF to casting, MgO and Al$_2$O$_3$ contents in inclusion increase. With the increase of feed calcium silicon line, the less its content increased, mainly with the calcium content in the LF refining process of further reducing and the reducing of the molten steel temperature, residual Mg and Al in molten steel continue to react with inclusions, make its content increase.

Figure 11 shows the changes of CaO and SiO$_2$ contents in the inclusions at different stages of the refining process with different Si-Ca wires addition amount. It can be seen from Figure 10 that the SiO$_2$ content in the inclusions increases significantly from 30% to about 80% before calcium addition. This is because the activity of SiO$_2$ in slag increase after silica sand addition, which is unfavorable to deoxidation by Si in Si-Mn-killed steel according to reaction [O] + [Si] = (SiO$_2$). In addition, the reaction between silica sand

particles that flowed into liquid steel and the inclusions in liquid steel are a possible route for the increase in the $SiO_2$ content in the inclusion. The change of CaO content in the inclusions was not obvious with the decrease of basicity of LF refining slag.

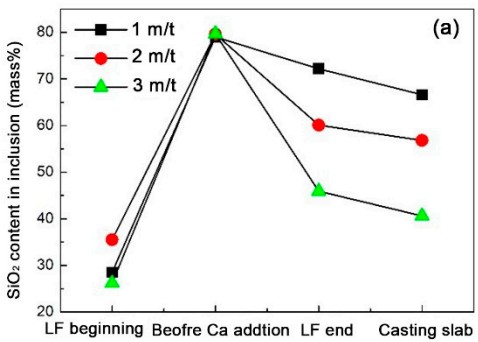 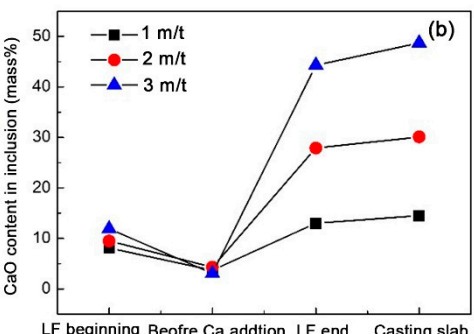

**Figure 11.** Influence of calcium addition amount on $SiO_2$ and CaO contents of inclusions at each stage. (**a**) $SiO_2$ content, (**b**) CaO content.

As can be seen from Figure 11a, after feeding Si-Ca wire in the LF refining process, the content of $SiO_2$ in the inclusions decreases with the significant increase of calcium content in the molten steel. With the increase of feeding wire amount, the content of $SiO_2$ in the inclusions decreases when end of LF, indicating that the calcium fed into the molten steel mainly reacts with $SiO_2$ in the inclusions. On the contrary, the content of CaO in the inclusion increases with the increase of wire feeding amount. When the wire feeding amount is 3 m/t, the content of CaO in the inclusion can reach more than 50%.

It can be seen from Figure 11b, there is an increase slowly in CaO in the inclusions from the end of LF to the casting slab with the decrease of the temperature of molten steel. While the content of $SiO_2$ in the inclusion decreases by about 5%.

Figures 12–14 show SEM images and EDS results of inclusions in casting slab in the case of different Si-Ca wires addition amount in LF. As can be seen from Figures 12–14, when the amount of Si-Ca wire feeding is 1 m/t, the EDS peak energy spectrum of $SiO_2$ in some inclusions is still high, and the content of CaO is very low. When the amount of Si-Ca wire feeding is 3 m/t, the peak energy spectrum of CaO is higher than that of $SiO_2$, and the melting point of inclusion will increase. The peak values of $SiO_2$ and CaO in the inclusions reached the best when the calcium wire feeding amount was 2 m/t, and the difference was not obvious when the calcium wire feeding amount was 1 m/t.

Figure 15 shows the compositions of oxide inclusions in liquid steel at the end of LF and in the casting slab on $CaO$-$SiO_2$-$MgO$-$Al_2O_3$ quaternary phase diagram. It can be seen from the Figure 15 that the melting point of the inclusions is less than or equal to 1300 °C for both the wire feed content of 1 m/t and 2 m/t. However, when the wire feed content is 2 m/t, the melting point of the inclusion is in the center of the low melting point region. When the wire feed content is 3 m/t, the melting point of the inclusion is about 1400 °C, and the melting point of the inclusion increases.

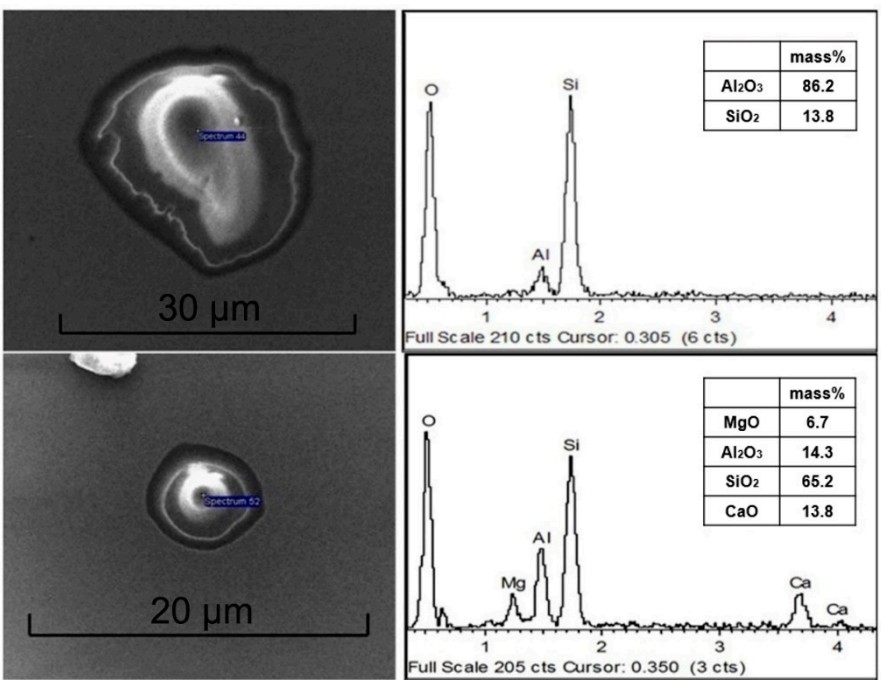

**Figure 12.** SEM images and EDS results of inclusions in casting slab in the case of calcium addition 1 m/t in LF.

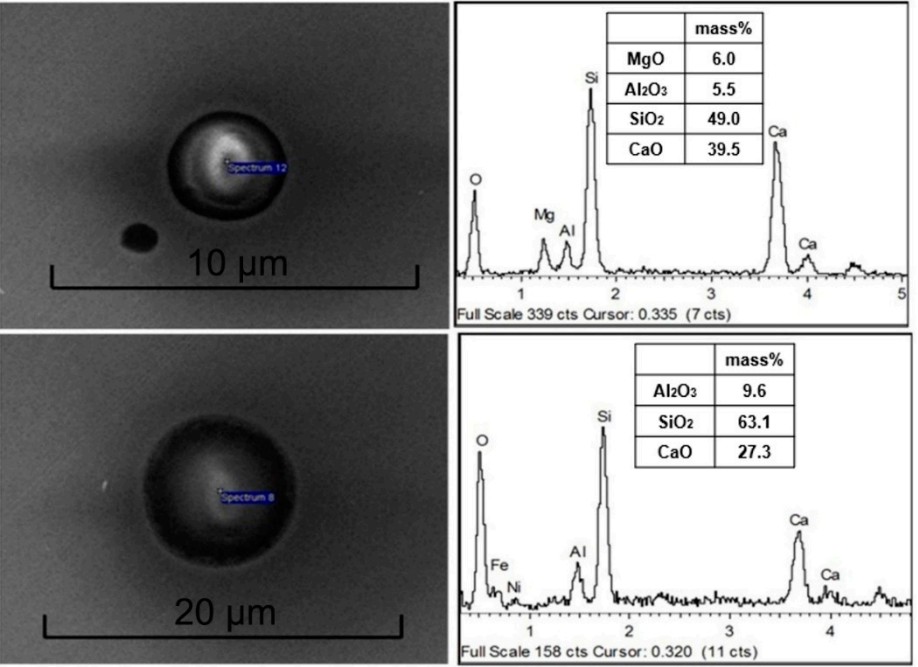

**Figure 13.** SEM images and EDS results of inclusions in casting slab in the case of calcium addition 2 m/t in LF.

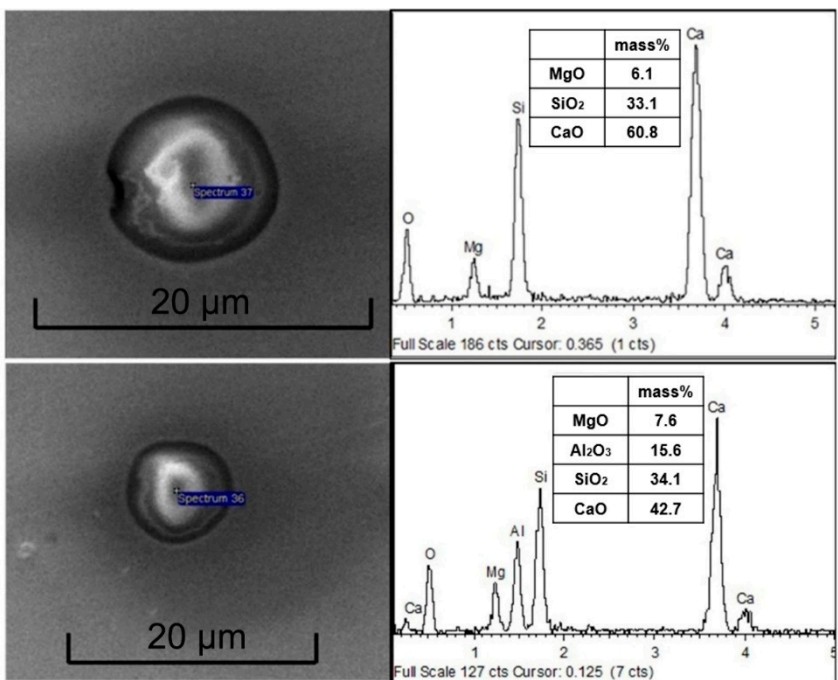

**Figure 14.** SEM images and EDS results of inclusions in casting slab in the case of calcium addition 3 m/t in LF.

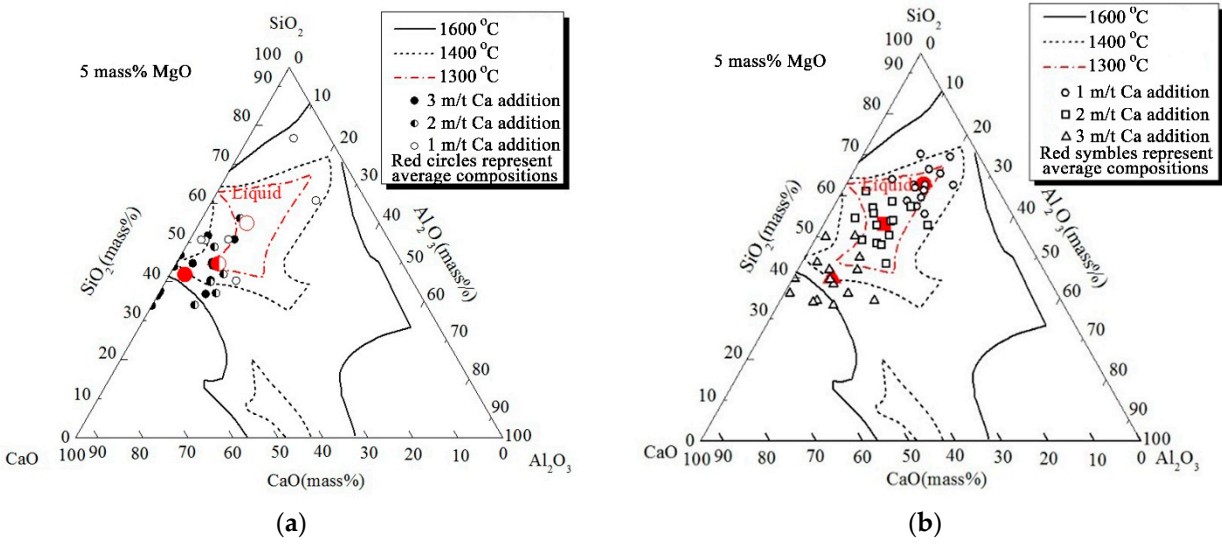

(**a**)                                  (**b**)

**Figure 15.** Compositions of oxide inclusions on CaO-SiO$_2$-Al$_2$O$_3$-5 mass% MgO diagram: (**a**) the samples taken from liquid steel at the end of LF, (**b**) the samples taken from casting slab.

### 3.2.2. Effect of Calcium Content on the Amount and Size of Inclusions

Figure 16 shows the number density of inclusions during LF refining with different calcium addition amount and in casting slab.

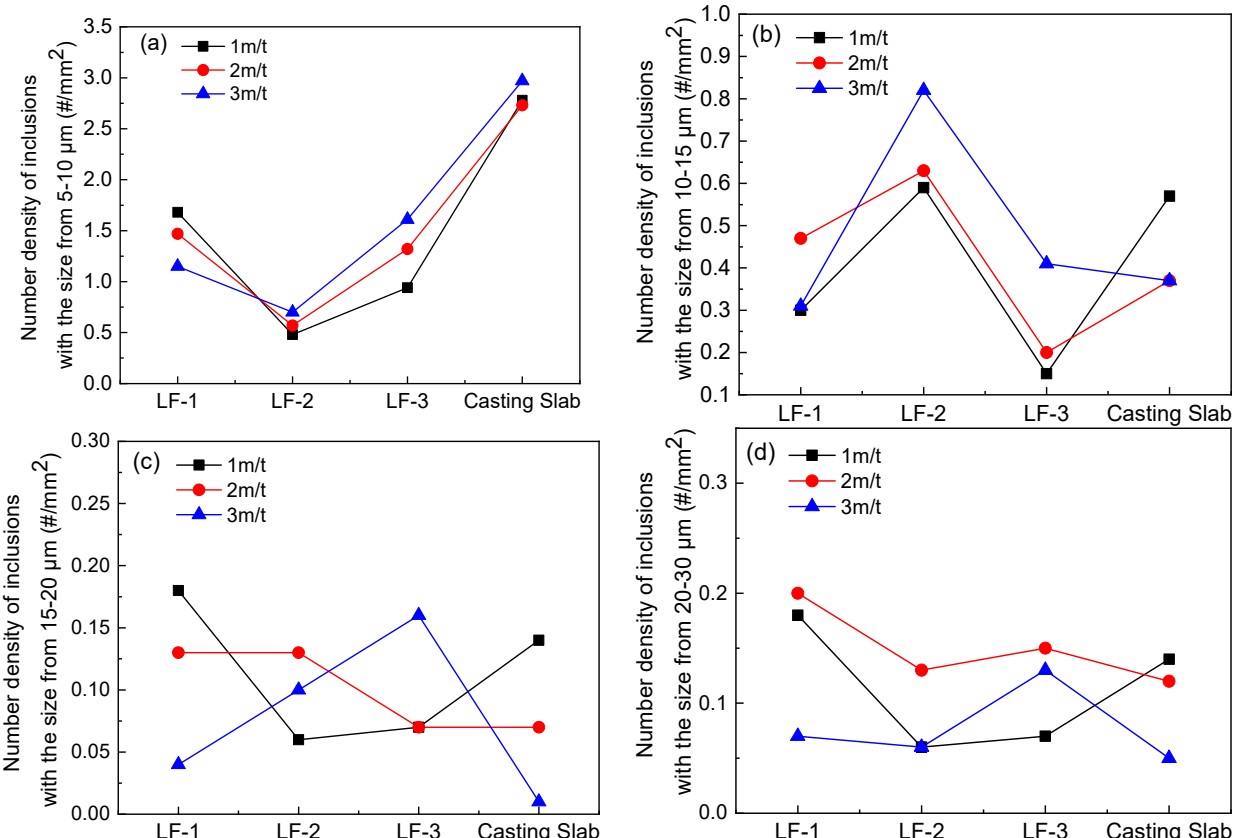

**Figure 16.** Number density of inclusions in liquid steel during LF refining and in casting slab when different Si-Ca addition amount is used in LF. Note: LF-1, LF-2, and LF-3 represent beginning of LF, before calcium addition at LF, and end of LF refining, respectively. (**a**) 5–10 μm, (**b**) 10-15 μm, (**c**) 15–20 μm, (**d**) 20–30 μm.

It can be seen from Figure 16a that after feeding Si-Ca wires into LF, part of calcium participates in deoxidation, which reduces the content of dissolved oxygen in molten steel and leads to an increase in the number of inclusions with a size of 5–10 μm. In the pouring process, as the temperature of the liquid steel decreased, the dissolved oxygen continued to react with elements Mg, Al, Si and Ca, forming small-size inclusions.

It can be seen from Figure 16b that the number of the inclusions with the size of 10–15 μm increases significantly from the beginning of LF to the stage before Si-Ca addition. There is no obvious change in the number of inclusions with the size of 10–15 μm from the stage of Si-Ca wires addition to casting slab.

The results in Figure 16c,d show that there are still the inclusions with the size larger than 15 μm in the casting slab, and the number density of inclusions has no obvious change from LF refining to casting slab.

### 3.2.3. Effect of Calcium Treatment on the Inclusions in Hot Rolled Plate

Table 9 shows the inclusion ratings in hot rolled coil after feeding different silicon calcium line in LF. As can be seen from that, the inclusions are mainly rated as B-type fine 0.5 and C-type fine 0.5~2.5 and some plastic inclusions have hard particles when the feeding amount of calcium wire is 1 m/t. When the feeding amount of calcium wire is 2 m/t, the inclusions show good plasticity, and the rating of inclusions is mainly C-type 1.5~2.5. Further increase the feeding amount of calcium wire up to 3 m/t, the rating of inclusion is mainly B-type 0.5 and C-type 1.0~2.0.

**Table 9.** Rating of the inclusion in hot rolled plate. (GB/T 10561-2005).

| Calcium Treatment | A-Type | B-Type | C-Type | D-Type |
|---|---|---|---|---|
| 1 m/t | 0.5 | 0 | 1.0~2.5 | 0.5 |
| 2 m/t | 0 | 0 | 1.0~2.5 | 0.5 |
| 3 m/t | 0 | 0.5 | 1.0~2.5 | 0.5 |

The experimental results of feeding Si-Ca wires in LF refining process show that,

(1) After feeding Si-Ca wires in LF refining process, where utilize the low basicity of slag, the contents of $SiO_2$, $Al_2O_3$ and CaO in the inclusions are reduced in varying degrees. When the feeding amount of calcium wire is 2 m/t, the content of CaO of inclusions in the casting billet is around 25%, and the melting point of inclusions enter the region of low melting point.

(2) The number of small-sized and medium-sized inclusions is increased after feeding Si-Ca wires, indicating that calcium participated in deoxidization after entering molten steel, but have no impact on the number and size of large inclusions in the casting slab.

(3) The rating of inclusions is mainly C-type in the hot rolled coil, after calcium treatment. However, a small amount of B-type inclusions will still remain, when the feeding amount of calcium is too large or insufficient. The optimal feeding amount is 2 m/t.

*3.3. Industrial Application of Inclusion Control Technology in 304 Stainless Steel Production*

3.3.1. Scheme of Plant Trials

The main reason for the defects after polishing of 304 stainless steel hot-rolled coil used in the clock industry is the presence of hard inclusions in the matrix. In order to meet the polishing performance, the melting points of inclusions should be low enough. The optimization scheme under industrial production is put forward, as seen in Table 10.

**Table 10.** Process optimization scheme for low-melting-point silicate inclusions control.

| | Original Process | Optimized Process | Aim of Optimization |
|---|---|---|---|
| AOD | using ordinary FeSi alloy | using ultra-purity FeSi alloy | reducing soluble Al in liquid steel |
| LF | without calcium addition | calcium addition 2 m/t before weak stirring | modify high-melting-temperature inclusions |

3.3.2. Industrial Production Process

The production process control is as follows.

**(1) AOD process**

In order to further reduce the Al content in liquid steel after AOD reduction, the ordinary FeSi alloy is replaced by ultra-purity Fe-Si alloy during AOD reduction. And, to further reduce the MgO content in slag after AOD reduction, the amount of light-burned dolomite was reduced from 4500 kg to 2500 kg during AOD melting.

Due to the addition of silica in the slag adjustment process of LF, the basicity of slag will be reduced properly. The sulfur in slag will return to liquid steel, resulting in the content of sulfur in liquid steel exceeding 50 ppm, as the basicity of slag decreases. In order to prevent the high sulfur in steel product, changing slag technology ('two-stage basicity control') was used at the reduction stage during AOD steelmaking, and the basicity of AOD after steel is controlled at the range of 2.0–2.2. The content of sulfur in liquid steel is ensured to be no more than 0.002% after double slag process in AOD.

The plant data at the end of AOD is summarized for 19 heats. After the tapping, the content of soluble Al in liquid steel is reduced from 0.0045% before process optimization of AOD to 0.0035%. Figure 17 shows the distribution of soluble Al content in steel after the

process optimization of AOD process. The content of sulfur is reduced from 0.005 mass% to 0.0020 mass%, as shown in Table 11.

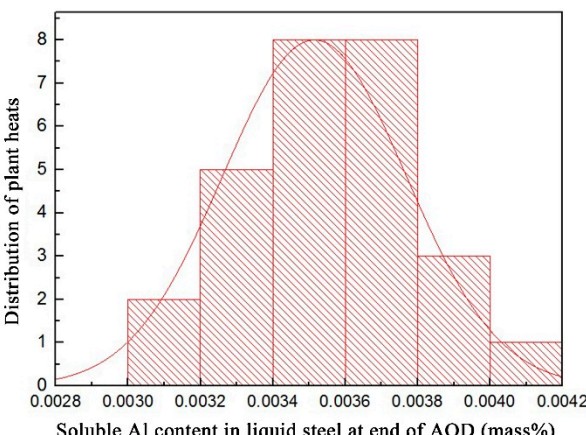

**Figure 17.** Content of soluble Al content in liquid steel at the end of AOD.

**Table 11.** The liquid steel composition before and after the process optimization (mass%).

|  | C | Si | Mn | P | S | Cr | Ni | Al | N |
|---|---|---|---|---|---|---|---|---|---|
| before process optimization | 0.035 | 0.45 | 1.1 | 0.035 | 0.0050 | 18.1 | 8.1 | 0.0061 | 0.055 |
| after process optimization | 0.035 | 0.45 | 1.1 | 0.035 | 0.0020 | 18.1 | 8.1 | 0.0035 | 0.055 |

Table 12 shows the change in slag composition at the end of AOD before and after process optimization. According to the table, the contents of $Al_2O_3$ and MgO in slag decrease from 2.53 mass% and 9.6 mass% before process optimization to 1.43% and 6.8%, respectively.

**Table 12.** Chemical composition of the slag before and after the process optimization (mass%).

|  | CaO | $SiO_2$ | $Al_2O_3$ | MgO | MnO | FeO | $Cr_2O_3$ | $CaF_2$ | R |
|---|---|---|---|---|---|---|---|---|---|
| before process optimization | 60.9 | 25.5 | 2.53 | 9.6 | 0.11 | 0.26 | 0.21 | 4.5 | 2.39 |
| after process optimization | 59.7 | 24.8 | 1.43 | 6.8 | 0.15 | 0.33 | 0.19 | 5.1 | 2.41 |

**(2)  LF process**

LF mainly deal with the accuracy in controlling slag basicity after slag adjustment. In this case, a facility for measuring thickness of slag is set in the process of slag picking station at the end of AOD, and the amount of silica sand added into LF can be determined by accurately measuring slag thickness. According to the large-scale production, the slag thickness of slag is generally controlled at 100–150 mm and the target is 120 mm at the begging of LF. 400 kg silica sand is added at the begging of LF, and after 10 min of power and slag melting, appropriate amount of silica sand is added to adjust lightly according to slag condition.

In order to ensure the effect of calcium treatment in LF process, the content of aluminum in the calcium alloy used in calcium treatment was analyzed. The results show that the content of aluminum in the Si-Ca wire alloy is 1.8%, and the content of aluminum in the pure calcium line is less than or equal to 0.1%. Consequently, the pure calcium line is used in LF process.

According to Stokes' theory, in order to make the inclusions fully float during the LF treatment process, time for the weak stirring and holding period in LF treatment were optimized and both of them are extended by more than 10 min, the number of large inclusions in the casting slab is significantly reduced by extending the time. Figure 18 shows the comparison of the size of the inclusions in the casting slab before and after extending the weak stirring and holding period for 10 min, it can be seen from Figure 18 that there are no inclusions larger than 20 μm in casting slab after prolonging the time.

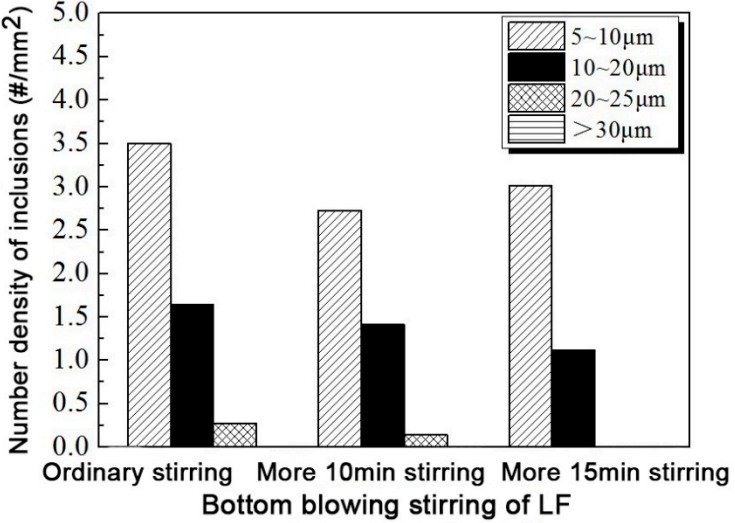

**Figure 18.** Size distribution of inclusions in casting slab after increasing weak stirring and holding period.

### 3.3.3. Effectiveness Analysis of Industrial Application

Hard inclusions in 304 stainless steel have been eliminated, and excellent deformability of inclusions by the application of the optimized AOD and LF process was achieved. Melting point of inclusions are lower than 1300 °C, and the inclusion rating of hot rolled plates is mainly C-fine-type 1.5~2.5 class, in which the typical chemical composition is 61.8% $SiO_2$-13.4% CaO-15.2% $Al_2O_3$-MgO 9.6%. The results of optimized AOD and LF process are as follows:

**(1) Cleanliness control of the casting slab**

The composition distribution of inclusions in the casting slab before and after optimizing the process is placed in the phase diagram, as shown in Figure 19. It can be seen that all the inclusions in the billet are located in low-melting-temperature region after optimizing the AOD-LF process.

**(2) Quality analysis of hot rolled plates after polishing**

The rating results of inclusions in hot rolled plates before and after the implementation of the process is shown in Table 13. It can be seen from the table that the inclusion rating of hot rolled plates after implementing the process is mainly fine C-type 1.5~2.5, with fewer A-type inclusions, the deformability of inclusions is fine, and the thickness of inclusions is 1 μm.

**Table 13.** Rating results of inclusions in hot rolled plates before and after the implementation of the process on phase diagram.

|  | Process | A-Type | B-Type | C-Type | D-Type |
|---|---|---|---|---|---|
| hot-rolled | Original process | / | 1.0 | 0.5~1.0 | 0.5 |
| plates (5 mm) | Experiment process | 1.0 | / | C1.0~2.5 | 0.5 |

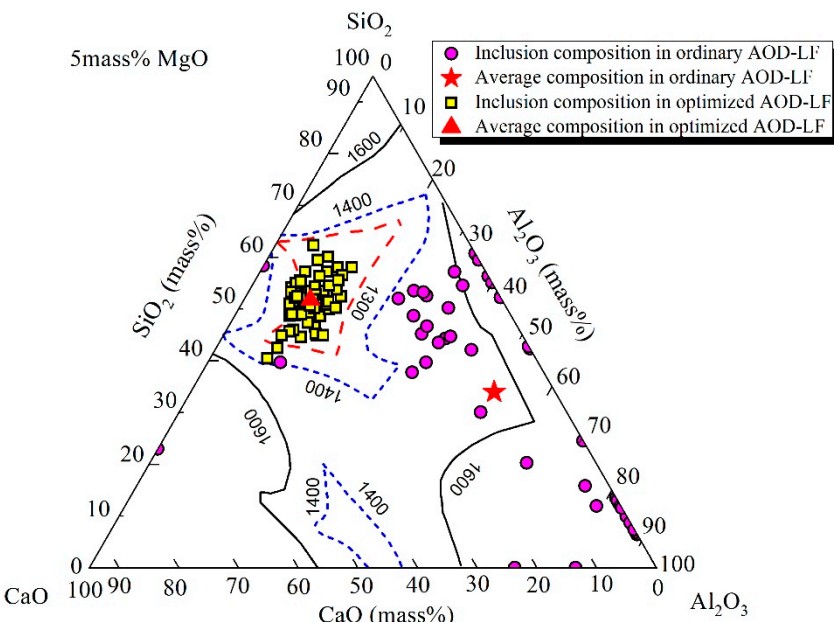

**Figure 19.** Composition distribution of inclusions in casting slab on $CaO-SiO_2-Al_2O_3-MgO$ phase diagram before and after optimization of AOD and LF process.

LF slag basicity is low in this process, leading to high sulfur content in the product, the content of sulfur in partial heat is above 0.004%. Therefore, in addition to silicate inclusions, a small number of MnS inclusions are found in hot rolled plates. Figure 20 shows the typical morphology and energy spectrum of MnS inclusions.

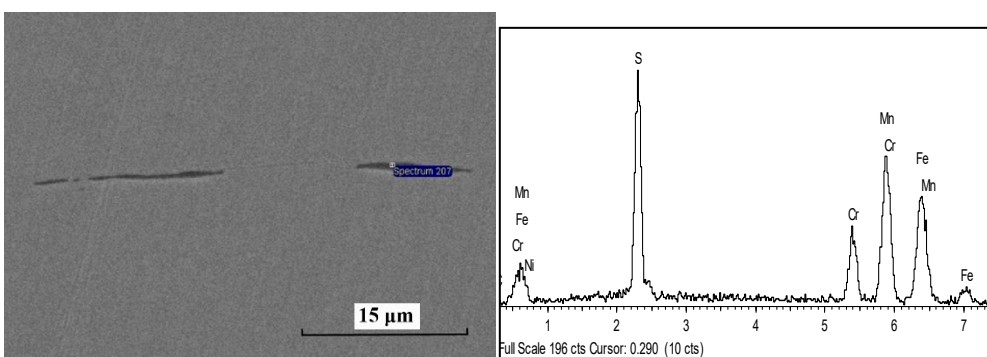

**Figure 20.** Typical MnS inclusions in hot rolled plates.

## 4. Conclusions

(1) With the increase of aluminum content in FeSi alloy, the contents of soluble aluminum in molten steel and $Al_2O_3$ in slag increase. When using ordinary FeSi alloy, the content of soluble aluminum in molten steel becomes higher than 0.005 mass%. The content of soluble aluminum in liquid steel could be limited to lower than 0.004 mass% when using ultra-purity FeSi alloy at the reduction stage of AOD.

(2) The contents of $SiO_2$, $Al_2O_3$ and CaO in inclusions decrease after Si-Ca wire addition. When the addition of Si-Ca wire is 2 m/t, inclusions are located in low-melting-temperature region, the inclusion rating of hot rolled plates is mainly C-type inclusion.

(3) Industrial application results show that after decreasing the content of soluble aluminum in molten steel and calcium treatment in LF refining, inclusions in 304 stainless steel could be controlled as low melting point silicate, the inclusion rating of hot rolled plates is mainly C-type with a small amount of A-type inclusions, surface polishing qualification rate increases from 17.8% to more than 88.7%.

**Author Contributions:** Conceptualization, J.Z. and C.S.; methodology, W.L. and S.W.; formal analysis, J.Z. and C.S.; investigation, J.Z.; writing—original draft preparation, J.Z. and C.S.; writing—review and editing, C.S., S.W. and Y.Z.; supervision, W.L.; funding acquisition, J.Z. All authors have read and agreed to the published version of the manuscript.

**Funding:** This research received no external funding.

**Institutional Review Board Statement:** Not applicable.

**Informed Consent Statement:** Not applicable.

**Data Availability Statement:** All data included in this study are available from the corresponding author on reasonable request.

**Conflicts of Interest:** The authors declare no conflict of interest.

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
