# Peer review of "Effect of FeSi Alloy Additions and Calcium Treatment on Non-Metallic Inclusions in 304 Stainless Steel during AOD and LF Refining Process"

_metals, doi:10.3390/met12081338_

Round 1

Reviewer 1 Report

Interesting article. Test results refer to industrial conditions. The research plan is extensive. The results of the article are obvious, the less aluminum is added, the smaller the amount of inclusions.

The use of the ASPEX MQA quality assessment system in research is justified.

The authors made an attempt to critically analyze the obtained results, which is an advantage of this work.

The test results are presented clearly.

There is no economic analysis for the presented research. It would be an interesting position in the article.

In Tables 1, 4 and 10, the Fe balance should be reported.

Author Response

There is no economic analysis for the presented research. It would be an interesting position in the article.

Response: The economic analysis is very multifactorial, and very compex. It can not be attributed to inclusion control. This is also not the emphasis of this study. 

In Tables 1, 4 and 10, the Fe balance should be reported.

Response: The Fe content is balance. Because the space of table, we did not list “balance”.

Reviewer 2 Report

Some noticing:

Raw 54: TiOx - TiOx

Raw 100-101: (in parentheses, state the Si content first)

Raw 140: How did you determine yield of Al% ?

Raw 151, raw 188: EDS or EDX spectra and no energy spectrums

Raw 168: How was calculated 5 mass%MgO?

Raw 189 and more: it is not evident from Figure 6, that no magnesia-alumina spinel inclusions were found in the furnace using ultra pure FeSi alloy.

Raw 239: Large amount of F element in the composition? What is the reason?

Results from Table 7 are not verbally evaluated.

Raw 273: from the table that.... The number of the table must be written.

Raw 282: from Figure 10 (a) - not only Figure 10.

Raw 284: SiO2 - SiO2

Raw 289: from Figure 10 (b)

Raw 311: Is it seen from Figure 15 that the melting point of the inclusions is less than or equal to 1300°C?

Raw 322: from Figure 16(a), When????

Raw 325: 16 (b) ???

Raw 327: there is no reference on literature 

Raw 333: feeding LF into Si-Ca wires???

Raw 384:  high sulfur of product?? What does it mean " double slag reduction"?

Raw 407: after the begging?

Raw 420-421: description is not in complete agreement with Figure 17.

Raw 426, 445: plastizication is used in connection with polymers

Raw 460: to Varying degrees?

Reviewer 3 Report

1.       Line 89-93

Although the authors mentioned the contents of Al in the FeSi alloy according to the grade, it is necessary to provide more detailed information that the contents of the other elements (such as Mg, Ca, Si, etc.) in each FeSi alloy.

Because the formation behavior of inclusions in the molten steel is significantly affected by Mg and Ca content in the ferroalloys.

So, I recommend the addition of a table listing composition of each FeSi.

2.       Experimental

To help understand the experimental process, it is recommended that the process sequence be illustrated graphically.

3.       Figure 1.

 What is the definition of aluminum yield?

Please indicate the definition of aluminum yield in the form of a formula.

4.       Figure 5.

In the case of a phase diagram containing 5% MgO, the axis should be expressed as 0~95%. We need to modify each axis of the triangle to represent 0-95% in Figure 5.

5.       Table 6.

The average inclusion composition at the time of AOD tapping shown in Figure 5 and the average inclusion composition at the start of LF shown in Table 6 are significantly different.

Since there are no events such as ferroalloy addition or slag addition between AOD tapping and LF beginning, there should be no significant difference between the inclusions at the start of LF and the inclusions at the time of AOD tapping.

Why is there a difference between them?

6.       Line 272-277

The Si activity in molten steel increases when the content of Si in the molten steel increases or when an element that interacts with Si is incresed. Since silica sand was added during the LF process, the pickup of Si is expected by the slag-molten steel reaction, but I think it is not enough to increase the SiO2 content in the inclusions to 80%.

In fact, as shown in Table 1, the difference in Si content in the AOD final and LF final is not large, so I think that the Si pickup is negligible.

Therefore, the author needs to check the change in the composition of the molten steel at each LF process, and it is necessary to review the composition of inclusions that are in equilibrium with the molten steel.

Additionally, it is necessary to examine whether the observed inclusions before Ca addition were formed by the physical influx of slag or silica sand. 

Reviewer 4 Report

Dear Authors,

You presented the manuscript of exclusively applied orientation. In your work, you study effects of refining additives and the effect of hot rolling on the concentration of unwanted inclusions and their shape after rolling. The manuscript is well-written, but needs considerable revision.

1. Please highlight the novelty of your work. What makes it different from many others and what contribution have you made to the subject area?

2. The presented reaction equations, lines 251 - 254 have nothing to do with the kinetics of the process. You give simple reaction equations based on thermodynamics.  In fact, here you have a complex multi-stage mechanism. 

3. There are no confidence intervals on the experimental curves.

4. The manuscript written sloppy: there are grammatical errors (for example, "untra" instead of "ultra"), table 2 is poorly readable (narrow intervals between columns), the presentation of mass percentages should be given in abbreviated form wt. %, the EDS results should be rounded up to tenths.

Good luck

Round 2

Reviewer 3 Report

Thank you for modifying the manuscript by reflecting the reviewers' opinions. However, in order to improve the quality of the manuscript, it is necessary to confirm the following contents more clearly.

Original review

6. Line 272-277 The Si activity in molten steel increases when the content of Si in the molten steel increases or when an element that interacts with Si is incresed. Since silica sand was added during the LF process, the pickup of Si is expected by the slag-molten steel reaction, but I think it is not enough to increase the SiO2 content in the inclusions to 80%.

Author Response: Silica sand was added during the LF process, the increase of the SiO2 content in the inclusions to 80% is originated from the pickup of soluble oxygen expected by the slag-molten steel reaction, not the pickup of Si. This is because the activity of SiO2 in slag increases after silica sand addition, which is unfavorable to deoxidation by Si in Si-Mn killed steel according to reaction [O]+[Si]=(SiO2). Under this condition, the SiO2 content in the inclusions is increased to 80%.

Reviewer Suggestions:

To certify the Author's argument, it is necessary for the thermodynamics consideration of the pick-up of soluble oxygen by the slag-metal reaction. That is, it should confirm the thermodynamics for the possibility of the (SiO2)slag = [Si] + 2[O] reaction and 2[O] + [Si] = (SiO2)inclusion reaction.

In the opinion of the reviewer, I think it is more persuasive to explain that silica sand particles flowed into the molten steel and reacted with the inclusions in the molten steel as a result, the SiO2 content in the inclusion increased.

Please check the above opinion.

Reviewer 4 Report

Dear authors,

You tried to answer my questions and comments. I am unfortunately unsatisfied with some of your comments. I strongly recommend rounding the SEM results to tenths, not hundredths.

Good luck

Author Response

Response: Thank you for your suggestions. We have revised SEM-EDS results to tenths.